# Understanding Feedback for Learners in Interprofessional Settings: A Scoping Review

**DOI:** 10.3390/ijerph191710732

**Published:** 2022-08-29

**Authors:** Varun Coelho, Andrew Scott, Elif Bilgic, Amy Keuhl, Matthew Sibbald

**Affiliations:** 1Faculty of Health Sciences, McMaster University, Hamilton, ON L8S 4L8, Canada; 2Department of Pediatrics, McMaster University, Hamilton, ON L8S 4L8, Canada; 3Department of Medicine, McMaster University, Hamilton, ON L8S 4L8, Canada

**Keywords:** feedback, interprofessional, education, training

## Abstract

Background: Interprofessional feedback is becoming increasingly emphasized within health professions’ training programs. The objective of this scoping review is to determine what is known about how learners perceive and interact with feedback in an interprofessional context for learning. Methods: A search strategy was developed and conducted in Ovid MEDLINE. Title and abstract screening were performed by two reviewers independently. Next, full texts of selected articles were reviewed by one reviewer to determine the articles included in the review. Data extraction was performed to determine the articles’ study population, methodologies and outcomes relevant to the research objective. Results: Our analysis of the relevant outcomes yielded four key concepts: (1) issues with the feedback process and the need for training; (2) the perception of feedback providers, affecting how the feedback is utilized; (3) professions of the feedback providers, affecting the feedback process; and (4) learners’ own attitude toward feedback, affecting the feedback process. Conclusions: The learner’s perception of interprofessional feedback can be an obstacle in the feedback process. Training around interprofessional feedback should be included as part of interprofessional programs. Research is needed to explore how to address barriers in feedback interaction that stem from misguided perceptions of feedback providers’ professions.

## 1. Introduction

The need for interprofessional education in healthcare has been growing for many years. In a 2010 publication, the World Health Organization recognized “interprofessional collaboration in education and practice as an innovative strategy that will play an important role in mitigating the global health workforce crisis” [1]. The Canadian Interprofessional Health Collaborative (CIHC) has defined interprofessional collaboration as the “process of developing and maintaining effective interprofessional working relationships with learners, practitioners, patients/clients/families and communities to enable optimal health outcomes” [2]. The CIHC National Interprofessional Competency Framework identifies six competency domains that are necessary for interprofessional collaboration: interprofessional communication, patient/client/family/community-centered care, role clarification, team functioning, collaborative leadership, and interprofessional conflict resolution [2]. Competency in these domains is essential for effective interprofessional collaboration and must be taught to health-profession learners. Interprofessional teams in healthcare can be quite diverse and can include learners, nurses, physicians, physical therapists, pharmacists, technicians, rehabilitation specialists, social workers and other professions that assist in the care of the patient. Different clinical scenarios can involve different fields working together to provide the best possible care to the patient. Interprofessional communication between these professions, particularly in delivering feedback and receiving feedback, has been recognized as an essential competency [3].

Feedback is defined as information about a person’s performance that serves to improve his or her capabilities and promote positive development [4,5]. The main characteristic of feedback is to provide learners with information about their observed performance compared to the desired performance [4]. Giving and receiving feedback is of vital importance in learning [4]. It is an essential component of education and provides learners an opportunity to gain valuable insight into their performance. Receiving constructive and timely feedback can help reduce the gap between actual and desired performance [5,6]. Importantly, feedback has been shown to be essential in health education in terms of learning and improving one’s skills [4,7]. Interprofessional teamwork, which includes feedback interactions, is recognized as important for quality patient care [8]. Although the literature highlights the importance of interprofessional feedback, there is less known about the process in which interprofessional learners give and receive feedback. Feedback delivery is now being recognized as not a simple flow of criticism and comments from teacher to learner, but rather a more complex process [5,9,10]. Understanding the various dimensions to giving and receiving feedback is of importance for interprofessional education, so that interprofessional learners can truly benefit from the feedback process.

A preliminary search of different databases, conducted on Ovid MEDLINE, showed very few reviews of the literature on how learners give and receive feedback in an interprofessional setting. Understanding what the literature says about interprofessional feedback is particularly important for educators. Knowing what makes feedback effective can assist educators seeking to teach health-profession learners in interprofessional settings. Scoping reviews provide a unique opportunity to analyze the existing literature to identify key concepts; gaps in the research; and types and sources of evidence to inform practice, policymaking and research [11]. Thus, this review’s objective was to understand what is known about how undergraduate learners give and receive feedback in an interprofessional context for learning.

## 2. Materials and Methods

This scoping review was guided by using the Arksey and O’Malley’s framework [11]. Our protocol was primarily formed in accordance with the Preferred Reporting Items for Systematic reviews and Meta-Analyses extension for Scoping Reviews (PRISMA-ScR) [12] and was revised by the research team. A scoping review was chosen as this study’s methodology, as scoping reviews provide the opportunity to map out the literature. Interprofessional education is a broad topic; therefore, a scoping review was chosen to identify what is known on this topic.

### 2.1. Review Question

What is known about how health professions’ learners give and receive feedback in an interprofessional context for learning?

### 2.2. Search Strategy

A search strategy was planned with the assistance of a health-sciences librarian. Keywords and Medical Subject Headings were identified: education, medical education, undergraduate medical education, feedback, feedback literacy, operations research, interprofessional, interprofessional relations and additional terms. The specific search algorithm used can be found in Appendix A, Table A1. A search of Ovid MEDLINE was conducted in October of 2021 with all of these terms. All relevant citations were uploaded into Covidence [13], where duplicate references were identified and removed.

### 2.3. Eligibility Criteria

We included all studies where empirical data were generated. This included studies with experimental, quasi-experimental, observational (e.g., prospective cohort) and qualitative (e.g., grounded theory) methodologies. Narrative or opinion-based studies were excluded. This scoping review focused on health professions’ learners in interprofessional settings. Specifically, these learners must be part of a profession that involves interactions with other professions. There were no restrictions on professions of the study participants as long as they were healthcare related and included in an interprofessional context. Works from the literature that included preceptors in the study population were included if learners were involved in its study population as well. Any works from the literature that did not include health professions’ learners was excluded. Studies with a central focus on feedback were included in this review. These included studies that analyze perceptions of feedback, the utility of feedback and the giving of feedback. Studies that had feedback as a secondary outcome, where it was not the central focus of the paper, were excluded. This scoping review analyzed papers published in English from 2001 to 2021 in any country. Geographic location was not a factor in inclusion. Additionally, this review was not limited by any racial, gender or socioeconomic demographics and considered all interprofessional learners.

### 2.4. Screening

Following the upload to Covidence, two independent reviewers screened articles by title and abstract to determine if they should be included in the full-text review. The reviewers met after their initial screening to resolve any conflicts. A full-text review of the remaining articles followed wherein the articles were assessed against the inclusion criteria. Reasons as to why any articles were excluded were recorded.

### 2.5. Data Extraction

Data were extracted from all papers selected for the scoping review, and this was performed by a single reviewer. A data-extraction spreadsheet was developed by reviewers and included data items with the following headings: “study details”, which included details on author(s), date and country; “study objective”; “study population”; “methodology”; and “relevant outcomes”. While no specific appraisal tool was used to assess selected articles, the quality of articles was assessed by recording limitations found in the studies. Outcomes were deemed relevant if they answered the review question. The data were analyzed by a single reviewer who filled in the data-extraction table. The completed table can be seen in Table A2 in Appendix A. After data extraction, the relevant outcomes of the papers were analyzed to find key concepts among the articles. This was performed by finding and recording common themes amongst the articles.

## 3. Results

The search of Ovid MEDLINE yielded 705 articles. After the first screening, 667 articles were excluded. This left 38 articles, two of which had reports that were unattainable. Therefore, 36 articles were assessed for eligibility through a full-text review. This resulted in 20 articles being excluded, as they did not adhere to the eligibility criteria. Specifically, 12 studies were excluded for wrong outcomes, 4 were excluded for wrong participants and 4 were excluded for wrong study design. This left 16 articles for data extraction. A summary of the process is shown in Figure 1.

### 3.1. Literature Characteristics

Of the 16 articles, 11 were published between 2016 to 2021. The other five articles were published between 2007 and 2013. All articles had more than one author, and they all took place in North America or Europe, except for one article which did not have a specific country of focus. The methodologies of the studies varied. Nine articles used qualitative approaches, six used quantitative approaches and one used mixed-method approaches. In terms of the study population, medical residents were a common study population, with five studies exclusively focusing on them. Additionally, three studies focused on medical/healthcare students; one study focused on dental students and another focused on veterinary medical students. The other articles (*n* = 5) contained multiple health professions in its study population. These professions include medical residents, physicians (supervisory physicians and general practitioners), nurses, nutritionists, psychologists, rehabilitation therapists, social workers, pharmacists, pharmacy students and medical students. Of these studies, residents were present in three articles, and nurses were present in three articles, as well. Specific details on articles’ details, population, methodology and outcomes can be found in Table A2 in Appendix A.

### 3.2. Key Concepts

#### 3.2.1. Issues with the Feedback Process and the Need for Training

Six studies identified a need for training on the feedback process to address issues in the giving and receiving of feedback. Issues related to giving feedback include feedback not being delivered on time, the overabundance of feedback evaluations and feedback being too vague [14]. One article showed that medical residents considered feedback from all other professions as suboptimal [15]. Health-professional learners indicated that feedback which promotes learners’ professional development needs to be timely, constructive, encouraging and focused on ways to improve [16]. Additionally, the process of giving feedback can be moderately challenging, as well, with one study giving a survey to interprofessional learners which asked them how challenging the process was of giving feedback to their interprofessional peers [17]. One study that had healthcare students undergo a feedback-literacy program had positive effects [18]. For example, students realized they could actively seek feedback instead of waiting for it and that they could engage in the feedback process and make plans of improvement. This relates to another article which showed that medical students’ own initiative led them to find feedback more instructive [19]. From these articles, it becomes clear that there is a need for training for health-professional learners in how to approach the feedback process.

#### 3.2.2. Perception of the Feedback Provider Affects How the Feedback Is Utilized

Seven articles suggested that a learner’s perception of the person providing the feedback affects how learners react to the feedback. In one study that analyzed learners’ perceptions of feedback regarding the learner–educator relationship, it found that attitudes, perceptions, relationships and teacher attributes affected feedback-seeking behavior [20]. Another study found that feedback-seeking behavior is influenced by the leadership styles of the feedback provider [21]. It found, through a review of the literature, that learners tended to seek feedback from their supervisor if they viewed them as a transformational leader. This supportive nature of teachers has also been shown in a separate article that gave a questionnaire to residents to assess predictor, mediator and outcome variables [22]. This article found that supportive leadership led residents to perceive more feedback benefits and fewer costs. Another study found that interpersonal factors, such as the relationship between the learner and the feedback provider, significantly influenced feedback-seeking behavior [23]. In another study, residents were found to actively contemplate the feedback they received based on their judgments of the feedback provider’s clinical credibility and their relationship with that feedback provider [24]. Similarly, another article found that identity and hierarchy is a significant theme in the feedback process, with social identities being the source of these interprofessional hierarchies [25]. In that article, residents valued interprofessional feedback and had a positive attitude toward it. However, regarding feedback from supervising physicians (an intra-professional identity to residents), the identities of “trainee” and “supervisor” became more highlighted as impeding receptiveness of feedback. This relates to another article where dental students were asked to evaluate their experience with feedback through a questionnaire and found that students rated faculty-led feedback higher than peer-led feedback [26]. These articles indicate that the way in which the learners view the feedback provider significantly influences the way that feedback is received.

#### 3.2.3. Professions of the Feedback Provider Affect the Feedback Process

The profession of the feedback provider can play a significant role in the feedback process. A study with a randomized control trial, with pediatric residents, highlighted the importance of multisource feedback [27]. Interestingly it showed that nurse ratings differed significantly than ratings from parents of the patients. This can indicate that the role or profession of the feedback provider matters in terms of the context of the feedback. Another concept that was found was that learners tend to value feedback more if it comes from their own profession. Three articles highlighted this theme, although one article seemingly contradicts it. One study found that there were significant interactions between the labelled profession of the feedback provider and the feedback recipient [28]. This means that nurses rated feedback that were labelled to be from nurses higher than feedback that was labelled to be from physicians, and vice versa. Similarly, two other studies showed that residents were found to value feedback from physicians more so than other health professionals [29] and were more likely to act on said feedback [15]. Residents preferred feedback from their in-group (physicians), as they felt physicians had better understanding of residents’ expectations and often focused on feedback that residents perceived as more valuable [15]. However, one article stated in its outcomes that the profession of the feedback provider had no main effect; there were no significant interactions between the professions of the feedback recipient and the feedback provider [17]. This article is contradictory to what the other articles were concluding; however, the article did find that the profession of the feedback recipient affected how feedback was rated, with some professions giving higher ratings than others. In conclusion, learners may value feedback from their own professions; however, more research may be needed to confirm this notion.

#### 3.2.4. Learners’ Own Attitude toward Feedback Can Affect the Feedback Process

A final concept that emerged from the data was the effect of the attitude of the learner. Three articles highlighted this concept. Two articles showed that learners with learning-goal orientations had positive interactions with feedback [21,23]. Learning-goal orientations, as described by the articles, is the desire to develop oneself by improving one’s skills and competence and mastering new situations [21,23]. One article found that there is higher value in feedback when oriented learners have learning goals because these learners view their abilities as something that can be improved over time [21]. These learners also view failure as a way to increase effort, making them less afraid of negative feedback. Another study found that residents who perceived benefits from feedback were found to report a higher frequency of feedback inquiry and monitoring (taking in self-relevant information from the environment by observation of others) [22]. These articles show that the attitude of the learner plays a prominent role in how they receive feedback.

## 4. Discussion

### 4.1. Summary of Main Findings

This scoping review focused on reviewing the scientific literature to discover what is known about how health-professional learners give and receive feedback in an interprofessional setting for learning. The literature reports used a variety of methodologies, with most using qualitative methods. Additionally, residents were found to be the most common study population. An analysis on the outcomes of the study yielded four key concepts: (1) issues with the feedback process and the need for training, (2) the perception of the feedback provider affects how feedback is utilized, (3) the professions of the feedback provider affect the feedback process and (4) the learners’ own attitude toward feedback can affect the feedback process.

### 4.2. Discussion of Findings

The key concepts bring to light many important elements on how learners give and receive feedback in the interprofessional environment. First, in regard to the issues in the feedback process, the findings show that training on the feedback process is needed. The articles from the first key concept indicated that feedback across professions can be suboptimal if it is not timely, specific, constructive and encouraging [14,15,16]. The article that used a literacy program for teaching feedback showed that these barriers can be partly addressed through education [17]. Additional evidence also suggests that improving the feedback literacy of students and staff could promote effective feedback practices and address some of the barriers [30,31]. This is important, as the type of feedback and the way it is given can have a significant impact on the feedback recipient. In particular, feedback has been shown to be more effective when it specifically focuses on a certain task and how to improve upon it, rather than feedback that just praises or punishes behavior [6]. Feedback, if not delivered properly, can even have negative effects on learners [9], thus highlighting the importance for improving it.

Additionally, the findings of the scoping review also show that the attitude of learners toward feedback can greatly affect how it is used. Learners with learning-goal orientations were mentioned in three articles of the scope [21,22,23]. Thus, teaching learners how to orient themselves in a way that is focused on improving oneself can potentially improve their competence. Therefore, it can be interpreted from the findings that feedback literacy needs to be implemented for both feedback providers and feedback recipients in order for feedback to be effective for learning.

Interprofessional teams can be quite diverse and include multiple professions collaborating to provide the best possible care to the patient. This diversity in medicine can be a potential strength in healthcare, as it allows the perspectives from multiple fields to be brought into consideration. However, one must understand the various group dynamics that come into play in an interprofessional environment. Perception of the feedback provider, including the perception of their profession, seems to play a significant role in how interprofessional feedback is used and interpreted. Studies showed that in-group and out-group biases can have important interactions on feedback in interprofessional settings. For instance, nurses will value feedback from nurses more so than feedback from physicians and vice versa [28]. These findings are not entirely surprising, as social-identity theory predicts that individuals will preferentially value perspectives from the group they identify with [32]. This can potentially be a negative aspect, as an interprofessional team can lose its inherent value by members of the team only valuing perspectives from their own profession. Group processes in medicine through the lens of social-identity theory can bring forth tensions in interprofessional teams which can affect the utility of feedback [33].

### 4.3. Limitations

The search of only one database, Ovid MEDLINE, is a limitation, as it may have restricted the number of articles that were found. Searching more databases may have provided more studies or a greater variety of study population. Additionally, the terms used for searching Ovid MEDLINE may not have been comprehensive enough and could have limited the articles found. Furthermore, all of the literature came from North America and Europe. This can potentially limit the scope of interprofessional education, as culture could play a role in interprofessional feedback. Additionally, there is a lack of confirmatory work in this scoping review. Having replication studies could help confirm the key concepts that were found.

## 5. Conclusions

Feedback among health-profession learners in an interprofessional context consists of many factors and is immensely important in health professions’ education, especially considering the growing need for interprofessional collaboration. This scoping review highlights important findings about how learners give and receive feedback in an interprofessional setting for learning. The review found that there is a need for interprofessional feedback training to not only address the issues in feedback but also to help learners develop a positive attitude toward the feedback process. More research needs to be performed to address learner’s perceptions of feedback providers and how to remove unwarranted biases toward out-group professions. While this scoping review includes a variety of works from the literature, more confirmatory work is needed to show how these concepts truly work in an interprofessional setting. Nevertheless, these findings contribute important information on the intricacies of feedback in interprofessional settings.

## Figures and Tables

**Figure 1 ijerph-19-10732-f001:**
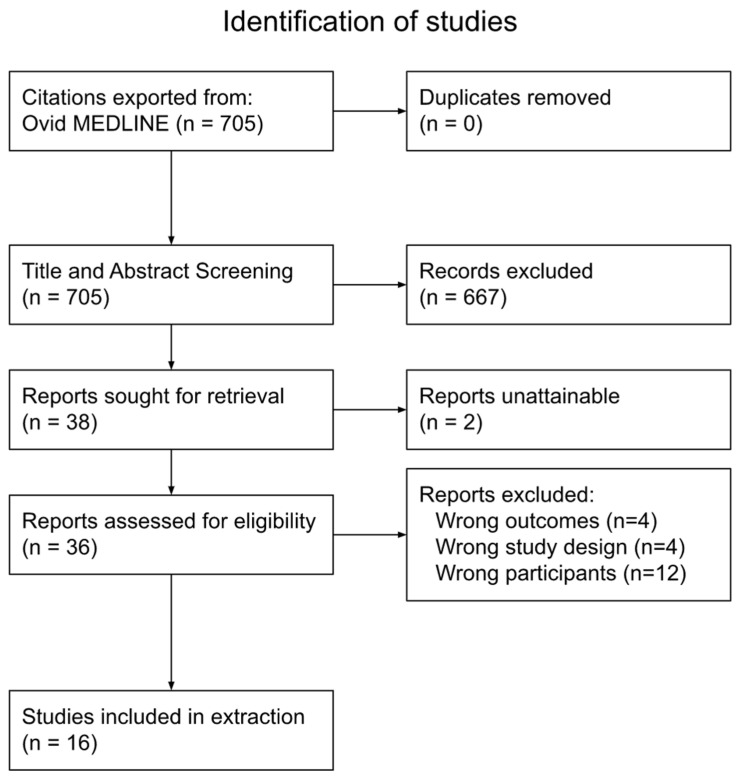
PRISMA-ScR diagram.

## Data Availability

No new datasets were created or analyzed in this study. Data sharing in not applicable to this article.

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
