# Peer review of "Understanding Feedback for Learners in Interprofessional Settings: A Scoping Review"

_ijerph, 2022, doi:10.3390/ijerph191710732_

Round 1
Reviewer 1 Report
Thank you for the opportunity to review this manuscript. The study is a scoping study of undergraduate health professionals interprofessional feedback, intended to map what is known about how health professions learners give and receive feedback in interprofessional learning context.
The aims and purpose of the study were clear, and the method well-articulated. The findings were articulated well, in particular the brief "implications" summary at the end of each concept made identifying the "so what" of the findings very easy. The discussion was succinct with specific calls to action that should guide educators and researchers in the future. The limitations were clearly outlined.
I would have liked some contextualisation of interprofessional teams. While the definitions were clear, the professional backgrounds of interprofessional teams differ depending on clinical context. Some commentary on the diversity of situations in which interprofessional teams work, and their membership might help the reader consider the transferability of the findings and conclusions of the study. The search terms were oriented towards "medical" education, which may also have limited the range of studies included, and I wondered if this was something the authors had recognised.
Reviewer 2 Report
I would like to thank the authors for their work.
This is an interesting paper, which aims to determine what is known about how learners perceive and interact with feedback in an interprofessional context for learning, with a scoping review design.
The introduction is clear and well developed, with a clear objective.
The methodology is robust and consistent with the PRISMA-scr method.
The results are clear and well organized, and the discussion in not speculative.
The main limitation, the choice to use only one database, is clearly declared.
The conclusion is supported by the results.
I don't have anything to add, and I consider this manuscript suitable for the publication.
Reviewer 3 Report
I have carefully read the article "Understanding feedback for learners in interprofessional settings: A scoping review" in which the authors use the Arksey and O'Malley's framework and PRISMA-ScR, to determine what is known about how learners perceive and interact with feedback in an interprofessional context for learning, in this regard I indicate my observations below.
Main concern
The authors need to follow the items contained in PRISMA-ScR properly; there are some elements that the authors do not adhere to, for example, protocol and registration, data items, critical appraisal of individual sources of evidence, among others.
Other aspects
The introduction is very brief, and the relevance of the study is not properly dimensioned, it is suggested that authors develop what feedback is, in terms that the authors use it, as well as its main characteristics; likewise, it is necessary to deepen into the characteristics of interprofessional environments and their importance in the educational context.
Line 70. The authors must define some of the search algorithms used; if there have been several, they can include them as supplementary material. Likewise, the authors need to define the period in which the search was carried out.
Line 77. It is recommended that points 2.3 and 2.3.2 be merged into one and start with types of sources that would immediately convey which studies were considered in the review.
Line 97. Point 2.3 is repeated, so it should be 2.4, and 2.4 should be 2.5 (Please review and adjust to the previous comment).
Table A1 in Appendix A. It would be highly recommended to include in the table the main limitations of the studies analyzed; this would allow readers to evaluate the articles more objectively.
In the results section, it is recommended to use one or more figures where the most critical aspects of the study are synthesized, systematized, or highlighted; otherwise, the presentation of the information may be monotonous for the reader.
The discussion is very brief, and in some parts, contains general information; it is recommended that the authors extend it emphasizing the importance of their study, example, lines 238-239: "Additional evidence show education can play a significant role in addressing the barriers of the feedback process [27, 28]". The authors do not specify what this additional evidence is and its significance in addressing the barriers of the feedback process; likewise, they point out: "This is important as the type of feedback and the way it is given can have a significant impact on the feedback recipient [4 ]." But they do not deepen or develop this idea.
Round 2
Reviewer 3 Report
I have reviewed the manuscript "Understanding feedback for learners in interprofessional settings: A scoping review," and the authors have incorporated most of the requested changes and addressed the evaluators' comments. In addition, the authors have increased the information in the table, which was relevant to analyzing the presented work with greater objectivity.